# Eco-Efficiency and Stock Market Volatility: Emerging Markets Analysis

**Alicia Fernanda Galindo-Manrique** [1], **Esteban Pérez-Calderón** [2,*] and **Martha del Pilar Rodríguez-García** [3]

[1] Accounting and Finance Academic Department, Instituto Tecnológico y de Estudios Superiores de Monterrey, Av. Eugenio Garza Sada, 2501 Sur Col. Tecnológico, Monterrey 64849, Mexico; alicia.galindo@tec.mx

[2] Accounting and Financial Economy Department, University of Extremadura, Av. Elvas, s/n, 06004 Badajoz, Spain

[3] Accounting and Administration Faculty, Universidad Autónoma de Nuevo León, San Nicolás de los Garza 66451, Mexico; martha.rodriguezgc@uanl.edu.mx

[*] Correspondence: estperez@unex.es

**Abstract:** Climate change, the accelerated industrialization of emerging countries, as well as the growing demand for transparency from stakeholders, are all factors that influence the environmental performance of companies. Thus, eco-efficient behavior can improve financial performance by increasing wealth generation and decreasing the volatility of listed financial assets. There is a lot of previous literature showing diverse results of the effect of eco-efficiency on corporate profitability, but this is not the case when we refer to risk. This study analyzes the relationship between eco-efficient behavior and the share price volatility of companies traded in emerging markets. For this purpose, a sample of 346 companies listed in 24 countries was studied for the period between 2010 and 2017. The results show a positive effect. Thus, the recommendation is that a clear commitment to eco-efficient investment can improve the environmental impact of companies, from the private, public, and institutional spheres.

**Keywords:** volatility; eco-efficiency; financial performance; emerging markets; panel data

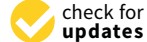



## 1. Introduction

Scarcity of natural resources and climate change pose a threat to the Earth and its inhabitants. The accelerated overheating of our planet has been caused mainly by $CO_2$ emissions generated by consuming fossil fuels. The cost of doing nothing outweighs the cost of repairing the harmful effects. Industries and financial markets must take on a greater leadership role in solving this problem (Cohen et al. 1997). Thus, companies must increase their commitment to the environment, but do so without forgetting their financial performance. This approach is part of the pillars that constitute corporate social responsibility (Carrol 1999; Taliento et al. 2019).

Shareholders and other stakeholders, in their need to understand the risks and opportunities associated with climate change, establish capital investment priorities and metrics to help them measure the benefit of environmentally friendly practices (Wahba 2008). The eco-efficiency theory takes on importance for all of these reasons. This theory aims to maximize economic value while minimizing environmental impact (Huppes and Ishikawa 2005; Porter and Reinhardt 2007; Nikolaou and Matrakoukas 2016). Currently, the link between environmental performance and financial performance has driven the demand and disclosure of environmental information by investors, and has played a crucial role in extra profit generation and risk reduction in financial markets (Henri and Journeault 2010; Closon et al. 2015; García-Sánchez and Araújo-Bernardo 2020).

Stock exchanges, driven by social and environmental responsibility initiatives, have generated green business models (Makower 2017). On the other hand, emerging countries have demonstrated their interest in the environment by incorporating factors associated

with the concept of sustainability into their financial markets. In the latter, in contrast to developed markets, such as Europe and the United States, emerging market economies have developed robust investment opportunities attracting capital flows from developed countries (Farooq 2015).

Global awareness of sustainability has boosted public policy initiatives that confirm the need to respond to a more informed consumer public and the promotion of environmental regulations (Banerjee and Solomon 2003). Member countries of the Organisation for Economic Co-operation and Development (OECD) have considered eco-innovation as one of the pillars of sustainable growth (OECD 2011). Governments in their role as overseers and regulators ensure economic prosperity and environmental balance (Henri and Journeault 2010). Thus, economic efficiency cannot be separated from environmental efficiency and it is necessary to be able to count on economic and environmental indicators that compare the evolution of regions or sectors. The above will lead to the development of effective policies locally and globally (Yang and Zhang 2018). The efficiency with which such policies and practices support the environment and measure society's response represent the fundamental pillars of economic growth, ensuring sustainable business and generating new sources of employment. Consequently, government policy actions on ecological issues can stimulate economic growth while combating environmental degradation, biodiversity loss, and unsustainable natural resource use (Costa 2021).

Few studies have focused their analysis on the effect of eco-efficiency on the risk sensitivity of financial markets, especially when it comes to emerging countries. Thus, one of the main contributions of this study is to provide further empirical evidence about the relationship between corporate eco-efficiency and the financial risk manifested by capital markets in emerging countries. For this purpose, 346 companies listed on stock exchanges were selected from three markets: Latin America, Europe and the Middle East, and Asia, with a total of 24 emerging countries. The selection of these countries was based on the Morgan Stanley Capital International (MSCI) Emerging Markets Sustainable Index. This research sample comprises cross-sectional and longitudinal data covering the period from 2010 to 2017.

The model presented in this study tests the theory of risk reduction associated with good behavior in greenhouse gas emissions, particularly $CO_2$ (Porter and Kramer 2006; Porter and Van der Linde 1995; Dhaliwal et al. 2011; Sharfman and Fernando 2008). This research presents the country effect and the effect of the industries denominated as penalizable. Such effects have also been previously studied, and we agree with the research conducted by Dhaliwal et al. (2011); Jo and Na (2012); Clarkson et al. (2004), among others.

The results of this study show evidence of a negative relationship between eco-efficiency and market risk, with a greater impact on companies with high volatility, mainly those in Asian countries and businesses that may be penalized from a Corporate Social Responsibility (CSR) perspective. On the other hand, European and Middle Eastern countries did not show such a relationship. This is in line with the regional studies conducted by Gottsman and Kessler (1998) and Jo and Na (2012).

## 2. Literature Review

The relationship between a company's social and environmental commitment and its financial performance has been a scientific concern since the 1970s. The shareholder approach, spearheaded by Friedman (2007), as well as the stakeholder approach (Freeman et al. 2004), drove the studies that are now the cornerstones of this topic. Researchers such as Orlitzky et al. (2003) and Margolis et al. (2009) have presented extensive evidence in the existing literature, concluding that the market rewards organizations' social and environmental responsibility initiatives.

More recently, the concept of economic and environmental efficiency, also known as eco-efficiency, emerged in the 1990s as a practical approach to gaining a better understanding of sustainability (Wagner and Schaltegger 2004). It was popularized by the World Business Council for Sustainable Development (Lehni and WBCSD 2000). On the other

hand, the OECD (2009) defines eco-efficiency as the efficient use of ecological resources to meet human needs. The theory of eco-efficiency proposed by Porter and Van der Porter and Van der Linde (1995) states that companies can maximize their efficiency while managing to reduce costs, create value, and minimize their environmental impact.

The indicators created to implement eco-efficiency are based on ratios related to the economic value of goods or services and their environmental impact (Huppes and Ishikawa 2005). Previous studies, such as those conducted by Berens and Cuny (1995), Dowell et al. (2000), Derwall et al. (2005), and Soyka and Feldman (2007), established that when companies effectively integrate eco-efficiency metrics into their operations, they create added value for their shareholders and decrease their risk profile in the stock markets. The values used in this research were carbon dioxide emissions, greenhouse gases, waste, energy use, and water consumption. In contrast, studies such as those by Belkaoui (1976), Freedman and Jaggi (1982), and Orlitzky et al. (2003) showed mixed or inconclusive results.

In the previous literature, we can find many works, such as those carried out by Shane and Spicer (1983), Wartick and Cochran (1985), Zhao et al. (2017), that focused mainly on the analysis of the effects of eco-efficient behaviors on economic-financial performance. More specifically, in terms of analyzing the effects of eco-efficiency, variables associated with economic profitability, such as competitive advantages generated, cost reduction, and sales increase, were used. From the variables used in the previous literature referring to financial profitability, we can highlight the following: the increase in net profit, Tobin's Q, or the revaluation of the company in the capital market (Lankoski 2007; Wagner 2015; Knox and Maklan 2004; Bendixen and Abratt 2007; La Rosa et al. 2018).

Spicer (1978) was among the first studies to analyze risk and environmental responsibility together. This author used companies in the paper-converting industry to measure the association between five financial growth variables. These comprised profitability, firm size, total risk, systematic risk, and price/earnings ratio. He compared these variables with the contamination index, and his results concluded that only systematic risk and the price/earnings ratio were statistically significant. In line with this finding, other studies have shown how externalities associated with environmental responsibility reduce the risk that is perceived by financial markets (Narver et al. 2004).

On the other hand, research carried out in emerging countries has exposed the constant regulation of governments to punish behavior that affects the environment. There is also pressure from consumers, since they are increasing their environmental awareness, according to Fernández-Gago and Nieto-Antolín (2004). This implies a clear trend towards the consumption of products and services generated by environmentally responsible companies. In the same sense, Charlo Molina and Clemente (2010) believe that there is a green awareness among investors and that this is part of the long-term investment criteria in capital markets.

Considering all the above, the objective of this study is to demonstrate whether eco-efficiency affects share price volatility in listed companies in emerging countries. The study started from similar previous research, such as that by Dowell et al. (2000), Hart and Ahuja (1996), Jo and Na (2012), and Guenster et al. (2011). Thus, the working hypothesis is as follows:

**Hypothesis 1.** *Eco-efficient companies are associated with lower share price volatility.*

The Jo and Na (2012) model was used as the main reference in determining whether eco-efficiency is negatively and significantly associated with share price volatility, and whether the fundamental variables have an impact on risk. Then, the model used is as follows:

$$Vol_{it} = \beta o + \beta_1 EE_{it} + \beta_2 NE_{it} + \beta_3 ROA_{it} + \beta_4\ IC_{it} + \beta_5 TA_{it} + \beta_6 PRSC_{it} + \beta_7 MAs_{it} + \beta_8 MResto_{it} + e_{it} \tag{1}$$

where:

- *it*, companies, and years that make up the data panel;
- $Vol_{it}$ is the standard deviation of stock returns;
- $EE_{it}$ is the eco-efficiency, $\frac{CO_2}{Sales}$ is $CO_2$ emissions over revenues, and the natural logarithm is used to standardize the results;
- $NE_{it}$ is indebtedness, calculated as $\frac{Total\ Debt}{Total\ Assets}$;
- $ROA_{it}$ is economic profitability, or a return on total assets;
- $CI_{it}$ is the investment level, calculated as $\frac{CAPEX}{Sales}$, with capital expenditures (CAPEX) as a function of total sales revenue;
- And $TA_{it}$ is size, or the natural logarithm of the total assets.

In the proposed model, the standard deviation was considered as the dependent variable ($Vol_{it}$). The financial theory states that total risk is composed of firm-specific risk and market risk. The total risk of an investment is measured by the variance or standard deviation of stock price returns (Ross 1976). Additionally, in previous works, this data was used as a reference for risk, as can be seen in the works of Derwall et al. (2005), Guenster et al. (2011), or Cohen et al. (1997).

The level of $CO_2$ emissions relative to sales ($EE_{it}$) was used as an independent variable, representing the level of eco-efficiency. This variable was calculated using the data set out in the environmental reports in the Environmental, Social and Governance (ESG) module of the Bloomberg data platform. Greenhouse gas emissions, measured through carbon dioxide, are the sum of the annual consumption of different energy sources: electricity, fuel, gas, heating, and air conditioning. As in previous studies, these emissions were standardized as compared to total sales (Dowell et al. 2000).

Firm size measured by the natural logarithm of total assets ($TA_{it}$) was used as a control variable. According to previous literature, firms with higher growth are more profitable and generate lower risk for investors (King and Lenox 2001). On the other hand, the financial structure of the firm ($NE_{it}$) is represented by the debt/total assets ratio. Research by Russo and Pogutz (2009) and Jo and Na (2012) already used this ratio as an indicator of debt level and as a control variable.

Russo and Pogutz (2009), Pérez-Calderón et al. (2011), and Alonso-Almeida et al. (2012) used this indicator to analyze the effect on social and environmental responsibility. To represent the company's level of investments ($CI_{it}$), the data was relativized in terms of annual sales, as was also done in the research of Jo and Na (2012).

Finally, the model included several dichotomous variables that discriminate against industries that may be penalized because their activity affects the ethics of investors. The Dummy penalization, from a CSR perspective (CSRP), classifies industries that are declared unethical or have a greater environmental impact. Thus, following Hong and Kacperczyk (2009), an industry was associated with unethical activities or high environmental impact if its main business is any of the following: alcohol, tobacco, gambling, nuclear energy, cement, oil, biotechnology, and weapons.

Finally, the study analyzed three main markets: America, Asia, and Europe and the Middle East group. This variable was intended to quantify the effect of the country grouped by its corresponding market. The *MAs* variable represents the Asian countries of China, Korea, Philippines, India, Indonesia, Malaysia, Pakistan, Russia, Taiwan, and Thailand. The variable *MRest* represents the European and Middle Eastern countries: Czech Republic, Greece, Hungary, Poland, Qatar, Turkey, United Arab Emirates, and the United Arab Emirates. It is important to note that the aforementioned markets are based on the American market, represented by Argentina, Brazil, Chile, Colombia, Mexico, and Peru.

## 3. Methodology

The panel data methodology was used to test the working hypothesis. This data analysis technique allows for more accurate inferences to be drawn, because it works with a greater number of degrees of freedom and reduces collinearity between the independent variables. Another relevant factor for the use of this model is the heterogeneity of the

observations concerning the study period in terms of the number of companies in each country (Nerlove and Balestra 1966; Arellano and Bover 1995).

It is important to note that the panel data model has two types of techniques available: fixed effects and random effects. Fixed effects assume that there is a characteristic of the dependent variable that is related to another independent variable. On the other hand, random effects assume that there is no correlation between the variables. This generates a serial correction between the unobserved effects and the independent variables, which assume a value of zero. However, this is not fulfilled in most cases. This implies inconsistency due to the variables omitted by the random-effects model. To test which of the two techniques best fit the panel data of the study, the Hausman (1978) test was performed.

Complementing this study, a quartile regression was used. Quartile regressions are useful when the conditional distribution does not have a standard shape, the tails are thicker and there are structural changes. They are also used when the principle of heteroscedasticity and the presence of outliers are met (Lv 2017).

## 4. Results

Table 1 shows the central tendency descriptive statistics results for the 346 companies in 24 emerging countries for the study period from 2010 to 2017 (see Appendices A and B).

**Table 1.** Variables and descriptive statistics.

|           | Vol     | EE       | NE       | ROA       | CI       | TA      |
| --------- | ------- | -------- | -------- | --------- | -------- | ------- |
| Mean      | 32.5930 | 4.4848   | 26.7243  | 5.8104    | 9.2253   | 12.3223 |
| Std. Dev. | 12.9419 | 3.4800   | 15.8726  | 7.75588   | 10.9753  | 2.6919  |
| Min.      | 8.101   | −7.8240  | 0        | −43.8523  | −39.18   | 5.7583  |
| Max.      | 156.009 | 14.2440  | 89.7431  | 120.812   | 190.7    | 19.5046 |

Following the methodology for the validation of the assumptions of normality, the Breusch and Pagan (1979) test was performed to determine the existence of heteroscedasticity. This test allowed us to analyze whether the estimated variance of the residuals of a regression depends on the values of the independent variables. The results showed significant levels in the Chi-square statistic and, therefore, the null hypothesis of homoscedasticity was rejected. This means that there is heteroscedasticity in the model, which implies that the variance of the errors is not constant, a main characteristic of financial assets in time series. The sample under study presented a phenomenon known as volatility accumulation, which means that there are lapses in which wide variations are shown for long periods, followed by an interval of relative tranquility. Thus, for stock price data, the basic assumption of the linear regression model is violated.

The correlation table shows the following relationship between the variables (see Table 2). Asset size company (*TA*) was negatively related to the eco-efficiency variable (*EE*) because of the economy of scale that large companies have in their production and sales processes. The indebtedness (*NE*) variable showed a direct relationship with eco-efficiency (*EE*), establishing that efficient indebtedness per unit of assets promotes sales efficiency by reducing emissions. The economic profitability (*ROA*) variable showed an indirect relationship, since higher levels of pollution per unit of sales have a negative effect on the return on assets. This establishes an inefficiency in the management and operation processes within the companies. Finally, the investment level (*CI*) variable maintained a positive relationship, representing the amount of investment in productive capital for the generation of sales. This suggests that eco-efficient companies acquire environmental competitiveness by incorporating efficient management in their productive and financial processes.

**Table 2.** Bivariate correlations.

| Variables | EE | TA | NE | ROA | CI |
|:---:|:---:|:---:|:---:|:---:|:---:|
| EE | 1.00 | | | | |
| TA | −0.69 *** | 1.00 | | | |
| NE | 0.26 *** | −0.14 *** | 1.00 | | |
| ROA | −0.17 *** | −0.02 ** | −0.41 *** | 1.00 | |
| CI | 0.08 ** | 0.05 | 0.10 *** | −0.04 | 1.00 |

*p*-value: *** < 0.01; ** < 0.05, * < 0.10.

In the correlation matrix presented in Table 2, significant values in the matrix indicate that the correlation is different from zero. This implies for our model that there is a significant relationship between the variables, even though the coefficients are small. To rule out the existence of multicollinearity in the model, it was necessary to perform a complimentary test. Multicollinearity is detected when there are high correlations between the predictor or independent variables in the model, and its presence can affect the regression results.

To confirm what was described in the correlation matrix, i.e., the non-existence of multicollinearity, the variance inflation factor (VIF) test was carried out (see Table 3).

**Table 3.** Results of the variance inflation factor test on the independent variables.

| Variable | VIF (1) | 1/VIF (2) |
|:---:|:---:|:---:|
| EE | 2.16 | 0.46 |
| TA | 2.07 | 0.48 |
| NE | 1.27 | 0.79 |
| ROA | 1.26 | 0.79 |
| CI | 1.04 | 0.96 |

As can be seen in Table 3, the VIF values were less than 10 and the mean was 1.56. The degree of tolerance, defined as 1/VIF, was used to determine the degree of collinearity. A tolerance value of 0.1 is equivalent to having a VIF of 10 and means that the variable can be considered as a linear combination of other independent variables, or that it is redundant. According to the data submitted, there was no collinearity between the independent variables.

The results of Model 1 eco-efficiency-risk are presented below. The model demonstrates the effect of eco-efficiency on stock price volatility in emerging markets. Table 4 describes the results of the model with the traditional panel data methodology and quantile regression for the 25th, 50th, and 75th percentile, respectively.

**Table 4.** Results of the regression model with panel data and quartiles.

| Var. | Panel | | 25 p | | 50 p | | 75 p | |
|:---:|:---:|:---:|:---:|:---:|:---:|:---:|:---:|:---:|
| EE | 0.3749 * | (1.86) | −0.1208 | (−1.11) | −0.158 | (−1.18) | 0.0265 * | (0.13) |
| NE | 0.1007 *** | (3.84) | −0.0034 | (−0.2) | 0.0165 | (0.79) | 0.0754 ** | (2.31) |
| ROA | −0.1213 *** | (3.58) | −0.2443 *** | (−7.81) | −0.3407 *** | (−8.83) | −0.3841 *** | (−6.34) |
| CI | −0.0627 ** | (2.41) | −0.0301 | (−1.41) | −0.0061 | (−0.23) | 0.0214 | (0.52) |
| TA | −0.2404 | (0.86) | −0.6337 *** | (−4.58) | −0.6867 *** | (−4.02) | −0.7911 ** | (−2.95) |
| PRSC | 2.0409 | (1.58) | 1.295 ** | (2.39) | 2.2752 *** | (3.41) | 3.4614 *** | (3.3) |
| Mas | 2.76084 ** | (2.01) | 2.6986 *** | (4.75) | 5.0324 *** | (7.19) | 6.5154 * | (5.93) |
| MRest | 1.9335 | (0.98) | 0.9456 | 1.21 | 2.1053 ** | (2.19) | 3.2665 ** | (2.16) |
| Cons. | 29.1636 | (7.25) | 33.0563 *** | (15.89) | 36.896 *** | (14.38) | 41.8145 *** | (10.38) |
| $R^2$ | 0.0918 | | 0.0336 | | 0.0541 | | 0.0725 | |

Note: values of the coefficient and the Z-statistic, respectively. *p*-value: *** < 0.01; ** < 0.05, * < 0.10.

The dependent variable in the model presented was represented by general volatility. It is important to note that the growth of the eco-efficiency variable indicates greater pollution. This is because this variable is composed of the amount of $CO_2$ emissions in its numerator and a denominator of the monetary units of sales. Companies pollute less by producing a lower amount of $CO_2$ emissions per unit of sales and, therefore, they are considered more eco-efficient. This model rejected the null hypothesis and, therefore, proved that there is an effect of eco-efficiency on volatility, which was significant at 1%. This is in line with studies by Jo and Na (2012), Dowell et al. (2000), Alvarez (2012), and Lv (2017).

In terms of the market discrimination effect, this was greater in Asia than in the Americas. The European and Middle Eastern countries did not represent a significant impact.

Asia's effect on the eco-efficiency-volatility relationship coincided with the results presented by the environmental strategy study developed by the World Bank (2018). Economic and population growth has generated severe negative environmental impacts in Asia. This report identified the triggering factors as being the lack of correction in environmental public policies and the exploitation of natural resources, as well as the accelerated growth in population and urbanization. These factors directly affect the Asian region's stock market.

On the other hand, the Latin American region has increased the integration of environmental and social responsibility reporting into the characteristics of its financial assets. Latin American countries have common denominator resources such as fertile soils, energy sources, and other underlying assets that can drive economic growth. Stock markets have been incorporating "green" investment strategies to attract the investing public.

Another important element is the return, represented by the *ROA*. In the model, the sign was negative, which indicates that the lower the risk, the lower the return. The same happened with the variables *TA* and *CI*. The results coincided with those obtained by Jo and Na (2012). The companies with an average volatility range that was lower and higher than the average, i.e., in the 25th, 50th, and 75th percentile, did not represent a significant level, so it was found that there was no effect of eco-efficiency on the different risk levels.

The results obtained were consistent, and thus, allow for the following arguments to be made. First, emission reduction activities provide a benefit for investment opportunities. Second, eco-efficiency strategies improve firm performance by creating long-term value by mitigating the risk perceived by investors. Third, from a CSR perspective, ethically punishable industries do not present significant effects for a low level of risk.

## 5. Conclusions

The contribution of this research is considered relevant for three main reasons.

First, given the scarcity of studies on the topic of eco-efficiency in emerging countries (Orsato et al. 2015), this research provides empirical evidence on the reduction of market risk as a function of pollution reduction in companies belonging to penalizable sectors.

The model under study is applicable to companies in other financial markets, since, according to the conceptual definition of eco-efficiency, it can be seen as an environmental performance indicator or as a sustainable development strategy (Koskela and Vehmas 2012). Eco-efficiency is achieved through three objectives: increasing the value of products and services, optimizing the use of resources, and reducing environmental impacts (Gottsman and Kessler 1998).

Furthermore, the functionality of the proposed model established four characteristics that can be implemented for any type of company, regardless of industry. These elements include environmental productivity, efficient production, environmental cost efficiency, and the implementation of environmental strategies that guarantee a cost-benefit balance (Huppes and Ishikawa 2005). In business management, the eco-efficiency model guarantees a reduction in the consumption of resources, a reduction in the impact on nature, and

an increase in the value of products and services for companies. This previous element means providing greater benefits to customers through functionality and flexibility in additional services focused on what the market really wants. In addition, this model has implications on risk management associated with the company's presence in capital markets.

Second, the use of volatility, determined by the standard deviation of stock price returns as a risk measure that captures the variability of financial assets, provides a decisive element for the incorporation of pollution reduction measures. This facilitates the elaboration of strategies within companies aimed at developing measures that benefit the environment, which, in turn, are viable long-term investment instruments (Russo and Pogutz 2009).

Thirdly, the study of risk, the effect of controversial sectors, and the country effect determine the following relationship: emerging economies, being in growth stages, tend to pollute more and this affects the volatility perceived by the markets. It is necessary to create public policies in emerging governments that focus on reducing damage to the environment and, at the same time, generate business opportunities that guarantee economic growth and the well-being of the population.

The results of the model showed the existence of the effect of eco-efficiency on market volatility. Companies pollute less by producing fewer $CO_2$ emissions per unit of sales and, therefore, decrease the variability in the share price.

The control variables showed the following behavior: leverage was significant at 1%. This result is similar to that obtained in research by Russo and Pogutz (2009) and Jo and Na (2012). The return on assets variable had a negative and significant effect on risk at 1% (Alonso-Almeida et al. 2012; Jo and Na 2012). Variable capital investments as compared to sales were significant at 5%, as was seen in the research conducted by Jo and Na (2012). The firm size variable was not significant, and this result was also presented in King and Lenox (2001) and Jo and Na (2012).

Complementing the analysis, the dichotomous variable of the penalizable industry was not significant in the overall panel data model, but it was significant in the quantile regression at the 75th percentile, indicating that companies in controversial sectors and with higher volatilities exert a greater effect on market risk. This result is consistent with that developed by Yoon et al. (2018).

On the other hand, the market effect was more significantly represented by the countries in the Asian and European and Middle Eastern regions than for the American market. The results obtained showed that the effect of the fundamental variable on market risk was greater in Asia than in the Americas. European and Middle Eastern countries did not represent a significant impact. Regional studies by Gottsman and Kessler (1998) and Jo and Na (2012) also demonstrated this effect.

According to the United Nations report (UN ESCAP 2009), Asian countries have demonstrated different levels of sustainability and economic development. The Asian region is known to have heterogeneous economies, characterized by different levels of economic development as a function of their $CO_2$ emissions. Although most Northeast Asian countries have shown steady progress in reducing their emissions between 1990 and 2010, the absolute value of $CO_2$ intensity has remained high in countries such as China and India.

The control variables of return on assets, capital investment as compared to sales, and company size also presented similar behaviors to previous models (Alonso-Almeida et al. 2012; Jo and Na 2012; Russo and Pogutz 2009).

To summarize the significance of the findings, the model developed facilitates a better understanding of the relationship observed between environmental performance and volatility in listed companies located in emerging countries. The observed effect relates companies with better eco-efficiency ratios to lower volatilities. At the same time, for the effect on companies that may be penalized for developing an unethical or environmentally harmful activity, only companies with the worst levels of eco-efficiency would be those that

carry the highest levels of risk. This provides additional information for companies and investors to help them make investment decisions that will generate value in the medium and long term.

The evolution of the corporate objective of creating economic benefits and protecting natural resources has shaped the generation of environmental metrics to economically measure the impact on corporations. Investors recognize that good environmental performance is an important source of business value by increasing long-term returns, improving market reputation, increasing efficiency in operations, enhancing innovation in processes, products, and services, as well as maintaining the loyalty of consumers and stakeholders in the community and markets.

The main objective of eco-efficiency is to provide governments with practical tools to measure their performance within the context of developing social and economic policies that guarantee environmental sustainability (UN ESCAP 2009).

Some future lines of research derived from the limitations of this study can be proposed. One of them would be the extension of the sample period to the time just before the crisis generated by COVID-19. Moreover, the analysis of the behavior of volatility during the pandemic, comparing the effects on the share prices of companies from emerging countries with those of companies from developed countries, would be of great interest. Other future lines of research that could emerge include the analysis of environmental metrics through eco-efficiency and the analysis of the economic growth of countries to measure macroeconomic variables that promote a benefit to humanity, whilst ensuring sustainable development at the same time. Another parameter to study is the role of financial performance at the microeconomic level and the effects on stock prices when reporting under *Sustainability* Accounting Standards Board (SASB) principles. Finally, the analysis of portfolios made up of polluting and non-polluting industries, and the effect on the financial risk of the companies, can also be studied. Finally, this research also proposes a sectoral and country analysis that can serve as a frame of reference to identify patterns of environmental risk and performance in the countries under study.

**Author Contributions:** Conceptualization, A.F.G.-M. and M.d.P.R.-G.; Data curation, A.F.G.-M.; Formal analysis, A.F.G.-M. and E.P.-C.; Investigation, E.P.-C. and M.d.P.R.-G.; Methodology, A.F.G.-M. and E.P.-C.; Software, A.F.G.-M. and M.d.P.R.-G.; Supervision, A.F.G.-M., E.P.-C., and M.d.P.R.-G.; Writing—original draft, A.F.G.-M. and M.d.P.R.-G.; Writing—review and editing, A.F.G.-M. and E.P.-C. All authors have read and agreed to the published version of the manuscript.

**Funding:** This research was funded by European Regional Development Fund, European Union "A way of making Europe", and by Council of Economy and Infrastructure, Regional Government of Extremadura (Spain), grant number GR18128.

**Institutional Review Board Statement:** Ethical review and approval were waived for this study.

**Informed Consent Statement:** Not applicable.

**Data Availability Statement:** Data available on request from the corresponding author.

**Conflicts of Interest:** The authors declare no conflict of interest.

## Appendix A. Frequencies of the Accounting Policies Regarding Economic Area and *Greenhouse Gas* Emissions

This table shows more information about the study sample used. This table represents the number of companies by country and economic zone. The data were compiled from the Bloomberg platform from World Bank (2018) and UNEP FI (2016) reports.

| No. | Country | No. Companies | Capitalization/GDP | $CO_2$ Emissions Per Capita |
|---|---|---|---|---|
| *AMERICA* | | *110* | | |
| *1* | Argentina | 4 | 17.05% | 4.75 |
| *2* | Brazil | 47 | 46.49% | 2.59 |
| *3* | Chile | 18 | 106.35% | 4.69 |
| *4* | Colombia | 12 | 38.63% | 1.76 |
| *5* | Mexico | 25 | 36.23% | 3.87 |
| *6* | Peru | 4 | 46.93% | 1.99 |
| *ASIA* | | *174* | | |
| 7 | China | 26 | 71.20% | 7.54 |
| 8 | Korea | 33 | 115.70% | 11.57 |
| 9 | Philippines | 10 | 92.60% | 1.06 |
| 10 | India | 19 | 88.00% | 1.73 |
| 11 | Indonesia | 10 | 51.30% | 1.82 |
| 12 | Malaysia | 7 | 144.80% | 8.03 |
| 13 | Pakistan | 5 | 33.00% | 0.9 |
| 14 | Russia | 2 | 39.50% | 11.86 |
| 15 | Taiwan | 41 | 50.00% | 13.2 |
| 16 | Thailand | 21 | 120.50% | 4.62 |
| *EUROPE AND THE MIDDLE EAST* | | *62* | | |
| 17 | South Africa | 16 | 352.80% | 8.98 |
| 18 | United Arab Emirates | 3 | 62.60% | 23.3 |
| 19 | Greece | 14 | 24.90% | 6.18 |
| 20 | Hungary | 4 | 22.60% | 4.27 |
| 21 | Poland | 6 | 38.30% | 7.52 |
| 22 | Qatar | 3 | 78.20% | 45.42 |
| 23 | Czech Republic | 3 | 17.40% | 9.17 |
| 24 | Turkey | 13 | 26.70% | 4.49 |
| *TOTAL* | | *346* | | |

## Appendix B. Frequencies of Penalizable and Non-Penalizable Companies by Sector

The penalty affecting the companies in the study sample from a CSR perspective is shown below. The following table classifies companies that are declared unethical or have a greater environmental impact. The data were compiled from the Bloomberg platform from World Bank (2018) and UNEP FI (2016) reports.

| Sector of Activity | Non-Penalizable | Penalizable | Total |
|---|---|---|---|
| Real estate | 4 | | 4 |
| Energy | | 36 | 36 |
| Finance | 67 | | 67 |
| Industrial | | 40 | 40 |
| Materials | | 48 | 48 |
| Non-commodity consumer products | 40 | | 40 |
| Consumer Staples | 27 | | 27 |
| Healthcare/Pharmaceuticals | | 7 | 9 |
| Communication services | 10 | | 10 |
| Telecommunication services | 14 | | 14 |
| Public utilities | | 25 | 25 |
| Information technology | 26 | | 26 |
| *TOTAL* | *190* | *156* | *346* |

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
