# Peer review of "Eco-Efficiency and Stock Market Volatility: Emerging Markets Analysis"

_admsci, doi:10.3390/admsci11020036_

Round 1

Reviewer 1 Report

The paper investigated the relationship between eco-efficient behavior and share price volatility of companies traded in emerging markets. The content of the paper corresponds to the chosen topic, but it is one of the shortest papers I have reviewed.

The paper is interesting but some aspect need to be improved.

In the Methodology section it is necessary to mention in the paper how the 346 companies were selected and to make a table with the number of companies in each country. It would be good to mention why the data stop in 2017.

For a better clarity of the paper, you can provide a figure explaining the relation between variables.

I believe that further discussion should be given as to whether the model would be applicable to companies in other financial markets.

Also can be included the implications of your research for management.

In the section Conclusions the limits of the paper must be added.

Author Response

The paper investigated the relationship between eco-efficient behavior and share price volatility of companies traded in emerging markets. The content of the paper corresponds to the chosen topic, but it is one of the shortest papers I have reviewed.

We appreciate your effort and your comments. For us, it is a great reward and satisfaction to see that our work can be of interest and contribute knowledge to the previous literature on the subject.

The paper is interesting but some aspect need to be improved.

In the Methodology section it is necessary to mention in the paper how the 346 companies were selected and to make a table with the number of companies in each country. It would be good to mention why the data stop in 2017.

As suggested by the reviewer, a reference to Appendix A has been included in the Results section. Appendix A appears at the end of the paper, which shows a more detailed description of the companies that make up the sample, with their data source (See Appendix A).

Additionally, the last part of the Conclusions section has been expanded to include the extension of the study period as a future line of research (see lines 378-388).

For a better clarity of the paper, you can provide a figure explaining the relation between variables.

Thank you for your suggestion. In response to the reviewer's comment, a paragraph has been included explaining the relationships observed between the variables in the model based on the correlations observed between them (see lines 235-246).

 The correlation table shows the following relationship between the variables (See Table 2). Asset size company (TA) is negatively related to the eco-efficiency variable (EE) due to the economy of scale that large companies have in their production and sales processes. The NE variable shows a direct relationship with eco-efficiency (EE) by establishing that efficient indebtedness per unit of assets promotes sales efficiency by reducing emissions. The ROA variable shows an indirect relationship since higher levels of pollution per unit of sales has a negative effect on the return on assets. This establishes an inefficiency in the management and operation processes within the companies. Finally, the CI variable maintains a positive relationship by representing the amount of investment in productive capital for the generation of sales. Eco-efficient companies acquire environmental competitiveness by incorporating efficient management in their productive and financial processes.

I believe that further discussion should be given as to whether the model would be applicable to companies in other financial markets.

Thank you for your comment. In the Conclusion section, the following paragraph was added (see lines 309-314).

 The model under study is applicable to companies in other financial markets, since, according to the conceptual definition of eco-efficiency, it be an environmental performance indicator or as a sustainable development strategy (Koskela and Vehmas 2012). Eco-efficiency is achieved through three objectives: increasing the value of its products and services, optimizing the use of resources, and reducing environmental impacts (Gottsman and Kessler 1998).

Also can be included the implications of your research for management.

Following the reviewer's recommendation, the following paragraph was added (See lines 315-324).

In the section Conclusions the limits of the paper must be added.

Thank you for your comment. Some limits of the study have been commented on at the end of the paper (See lines 378-392).

Reviewer 2 Report

Many thanks for the opportunity of reading and commenting your paper. It was very interesting and actual fulfilling a literature gap. 

Some comments will be raised with the purpose of improving the overall quality of the document, which I expect being taken into consideration. 

In the title shouldn't it be a mention to Stock Market Volatility? It is not obvious for readers what to expect... please consider the suggestion.

At first I feel that there is an overlooked perspective of eco-efficiency that needs to be further explored despite not being the core of the paper - the role of Government policy actions in the promotion of green practices, as it conditions firm decision making processes. In this vein, I suggest exploring a recent paper from resources: Costa, Joana. 2021. "Carrots or Sticks: Which Policies Matter the Most in Sustainable Resource Management?" Resources 10, no. 2: 12. https://doi.org/10.3390/resources10020012

As the authors know well, taxes will force immediate reaction, but, incentives can be neglected. 

I doubt on the formulation of the first hypothesis - if you mention "Emerging Economies" this means that you have a different expectation for other types of countries? - if so there is a need for further explanation. If it is just an application it should not be in the hypothesis. [I believe that it is the first]

Around line 182 perhaps there should be an econometric reference (e.g. Baltagi), the Nervole seems to bedated - Wooldridge or Arellano have more recent approaches for the case.

The reader is not informed about the data source. Empirical studies aim to be replicable, however, nobody is informed how the data was collected, nor where it comes from. The reduces readers' trust. What are the firms involved? ( at least a general idea needs to be given - exploratory analysis)

Correlation matrices commonly appear with only one half.

Table 4 appears with the wrong format in my version. The significance *** should go along with the coefficient not the s.e.

In the result section I would like to see more comparison with former studies and the implications. 

Lastly, and given the multidimensional scope of the aprraisal there is a need for creating a sub-section exploring contributions (theoretical and practical), recommendations (for practitioners, investors and policy makers)and the study limitations - highlighting the aftermath of the financial crisis could be of use.

Best of luck with the research!

Author Response

Many thanks for the opportunity of reading and commenting your paper. It was very interesting and actual fulfilling a literature gap. 

Some comments will be raised with the purpose of improving the overall quality of the document, which I expect being taken into consideration. 

We appreciate your effort and your comments. For us, it is a great reward and satisfaction to see that our work can be of interest and contribute knowledge to the previous literature on the subject.

In response to the reviewer's assessment of the level of English, the translation has been revised to improve some expressions and grammatical structures.

In the title shouldn't it be a mention to Stock Market Volatility? It is not obvious for readers what to expect... please consider the suggestion.

As can be seen in the title, there is already a mention of volatility in emerging markets. However, following the reviewer's suggestion, the title has been modified to make it more obvious to the reader.

At first I feel that there is an overlooked perspective of eco-efficiency that needs to be further explored despite not being the core of the paper - the role of Government policy actions in the promotion of green practices, as it conditions firm decision making processes. In this vein, I suggest exploring a recent paper from resources: Costa, Joana. 2021. "Carrots or Sticks: Which Policies Matter the Most in Sustainable Resource Management?" Resources 10, no. 2: 12. https://doi.org/10.3390/resources10020012

As the authors know well, taxes will force immediate reaction, but, incentives can be neglected. 

Thank you very much for the suggestion. To respond to the reviewer's comment, the following paragraph has been added to the Introduction section (see lines 50-64).

“Global awareness of sustainability has boosted public policy initiatives that confirm the need to respond to a more informed consumer public and the promotion of environmental regulations (Banerjee and Solomon 2003). Member countries of the Organisation for Economic Co-operation and Development (OECD) have considered eco-innovation as one of the pillars of sustainable growth (OECD 2011). Governments in their role as overseers and regulators ensure economic prosperity and environmental balance (Henri and Journeault 2010). So, economic efficiency cannot be separated from environmental efficiency and it is necessary to be able to count on economic and environmental indicators that compare the evolution of regions or sectors. The above will lead to the development of effective policies locally and globally (Yang and Zhang 2018). The efficiency with which such policies and practices support the environment and measure society's response represent fundamental pillars of economic growth, ensuring sustainable business and generating new sources of employment. Consequently, government policy actions on ecological issues stimulate economic growth while combating environmental degradation, biodiversity loss, and unsustainable natural resource use (Costa 2021).”

I doubt on the formulation of the first hypothesis - if you mention "Emerging Economies" this means that you have a different expectation for other types of countries? - if so there is a need for further explanation. If it is just an application it should not be in the hypothesis. [I believe that it is the first]

Thank you for your comment. The hypothesis has been reformulated in response to the reviewer's suggestion (See line 141).

Around line 182 perhaps there should be an econometric reference (e.g. Baltagi), the Nervole seems to bedated - Wooldridge or Arellano have more recent approaches for the case.

Thank you for your comment. A new previous literature reference has been included as recommended by the reviewer (See line 201).

The reader is not informed about the data source. Empirical studies aim to be replicable, however, nobody is informed how the data was collected, nor where it comes from. The reduces readers' trust. What are the firms involved? ( at least a general idea needs to be given - exploratory analysis)

As suggested by the reviewer, a reference to Appendix A has been included in the Results section. Appendix A, which shows a more detailed description of the companies and the data source, appears at the end of the document. Regarding the comment as to why the sample stops in 2017 the reason is that the study is the result of a doctoral dissertation and has not been updated. The authors appreciate the comment, and the suggestion has been considered as a future line of research.

Correlation matrices commonly appear with only one half.

Thank you for your comment. The proposed correction has been made (See Table 2).

Table 4 appears with the wrong format in my version. The significance *** should go along with the coefficient not the s.e.

Thank you for your comment. The proposed correction has been made (See Table 4).

In the result section I would like to see more comparison with former studies and the implications. Lastly, and given the multidimensional scope of the aprraisal there is a need for creating a sub-section exploring contributions (theoretical and practical), recommendations (for practitioners, investors and policymakers), and the study limitations - highlighting the aftermath of the financial crisis could be of use.

We appreciate the reviewer's suggestion. The introduction and conclusions sections have been expanded and more references to previous studies have been added to better contextualize and interpret our results.

Reviewer 3 Report

This paper has an interesting idea on how to study the effect of eco-efficiency behavior on volatility in emerging markets. The paper has a very good structure and is exemplarily short and to the point. 

The main contribution of the paper is to focus on the relationship between the eco-efficient behavior of companies and their financial performance in a geographic area in continuous economic growth.

Introduction and literature review are concise and adequate. Perhaps I would have included the presentation of the Jo and Na (2012) model in the methodology section. However, it is true this model is adequate for the work carried out. 

The results are correctly presented and the conclusions provided are relevant. 

Author Response

We appreciate your effort and your comments. For us, it is a great reward and satisfaction to see that our work can be of interest and contribute knowledge to the previous literature on the subject.

Reviewer 4 Report

Generally it is a good paper. The subject of the research is interesting, up-to date and worth scientific examination, considering one of the aspects of risk in a sustainable economy. The design of the study, structure of the paper, logic, state of the art analysis, hypothesis development, methods of testing it, conclusions are prepared correctly, according to the rules of scientific work. The aim of the work is clear and, in my oppinion, it was fully achieved with the use of proper methods. On a sample of 346 companies it was proved that eco-efficient companies are associated with lower share price volatility in emerging countries.

Still, I found few parts demanding further technical explanations:

Lines 205-210 - "...the null hypothesis of homoscedasticity was rejected", on a basis of Breusch-Pagan test. This means heteroscedasticity of the model. Please point out what are the implications of this fact for the model.

Table 2 - some of the examined correlations are statistically significant, so it is not true that "relatively small correlation values are observed" (line 215) and multicollinearity is present in some degree. Again, please explain what it means to the model.

Overall, research provides empirical evidence on the reduction of market risk as a function of pollution reduction in companies belonging to penalizable sectors. This is an important statement for the sustainable economy.

Author Response

We appreciate your effort and your comments. For us, it is a great reward and satisfaction to see that our work can be of interest and contribute knowledge to the previous literature on the subject.

Still, I found few parts demanding further technical explanations:

Lines 205-210 - "...the null hypothesis of homoscedasticity was rejected", on a basis of Breusch-Pagan test. This means heteroscedasticity of the model. Please point out what are the implications of this fact for the model.

Thank you for the recommendation. The paragraph has been completed with the following (See lines 221-231):

 “Following the methodology for the validation of the assumptions of normality, the Breusch and Pagan's (1979) test was performed to determine the existence of heteroscedasticity. This test allows us to analyze whether the estimated variance of the residuals of a regression depends on the values of the independent variables. The result showed significant levels in the Chi-square statistic and, therefore, the null hypothesis of homoscedasticity was rejected. This means that there is heteroscedasticity in the model, which implies that the variance of the errors is not constant, as one main characteristic of financial assets in time series. The sample under study presents a phenomenon known as volatility accumulation, which means that there are lapses in which wide variations are shown for long periods, followed by an interval of relative tranquility. For stock price data, the basic assumption of the linear regression model is violated.”

Table 2 - some of the examined correlations are statistically significant, so it is not true that "relatively small correlation values are observed" (line 215) and multicollinearity is present in some degree. Again, please explain what it means to the model.

Thank you for the recommendation. The paragraph has been completed with the following (See lines 235-241):

The above implies for our model that there is a significant relationship between the variables, even though the coefficients are small. To rule out the existence of multicollinearity in the model it is necessary to perform a complimentary test. Multicollinearity is detected when there are high correlations between predictor or independent variables in the model and its presence can affect the regression results.

Round 2

Reviewer 1 Report

The paper is improved, but there are minor aspect to be clarified.

I consider that more details about the sample should be specified in the paper, the appendix shows the distribution by countries and the number of companies in each country. Apart from the fact that the shares of these companies were very volatile, what other common characteristics do these companies have? Perhaps the field of activity, the impact of external factors or other factors have led to their volatility.

Author Response

We appreciate the reviewer's comment. Thank you very much for your dedication to improve our work. We have responded to your recommendation. Two new columns have been added to the table in Appendix A, thus increasing the detail of the sample of companies and their countries. Also, a new Appendix has been added with a new table showing the representation of companies by sector of activity and whether or not they belong to the group of companies that may be penalized to a greater extent in the markets for ethical, moral, or environmental issues (see Appendix B).

Reviewer 2 Report

Many thanks to the authors for providing an improved version of the article. The new manuscript encompasses, in general, the updates demanded.

I would like the authors to re-read the document as there are some typos and slight adjustments to be made before publishing.

In the tables some minor details are still missing: Table 1 has several mistakes - please correct them: "Statics", "St. Desv.", source. Table 4 - what is in parenthesis? must be mentioned. please remove the source in all of them.

In the conclusion section there is a need for one or two additional paragraphs enlightening the reader about the importance of your findings. Think like: why does it worth to consider your results when taking decisions? What is your final contribution?

Many thanks,

Good luck

Author Response

We appreciate the reviewer's comment. Thank you very much for your dedication to improving our paper.

In the tables some minor details are still missing: Table 1 has several mistakes - please correct them: "Statics", "St. Desv.", source. Table 4 - what is in parenthesis? must be mentioned. please remove the source in all of them.

The authors are grateful for the suggested corrections and have updated the paper in response to these comments.

In the conclusion section, there is a need for one or two additional paragraphs enlightening the reader about the importance of your findings. Think like: why does it worth to consider your results when taking decisions? What is your final contribution?

The authors are grateful for the reviewer's comments. To address this recommendation, the following paragraph has been added to the last part of the conclusions section.

To summarize the significance of the findings, the model developed facilitates a better understanding of the relationship observed between environmental performance and volatility in listed companies located in emerging countries. The observed effect relates companies with better eco-efficiency ratios to lower volatilities. At the same time, as for the effect on companies that may be penalized for developing an unethical or environmentally harmful activity, only companies with the worst levels of eco-efficiency would be those that carry the highest levels of risk. This generates additional information for companies and investors to help them make investment decisions that will generate value in the medium and long term.